# Long-Term Outcomes at Skeletal Maturity of Combined Pelvic and Femoral Osteotomy for the Treatment of Legg–Calve–Perthes Disease

**DOI:** 10.3390/jcm12175718

**Published:** 2023-09-01

**Authors:** Christina M. Regan, Alvin W. Su, Anthony A. Stans, Todd A. Milbrandt, A. Noelle Larson, William J. Shaughnessy, Emmanouil Grigoriou

**Affiliations:** 1Department of Orthopedic Surgery, Mayo Clinic, Rochester, MN 55905, USA; regan.christina@mayo.edu (C.M.R.); stans.anthony@mayo.edu (A.A.S.); milbrandt.todd@mayo.edu (T.A.M.); shaughnessy.william@mayo.edu (W.J.S.); grigoriou.emmanouil@mayo.edu (E.G.); 2Department of Orthopedic Surgery, Nemours (duPont) Children’s Health, Delaware Valley, Wilmington, DE 19803, USA

**Keywords:** Legg–Calve–Perthes disease, osteotomy, hip joint deformity, osteonecrosis of femoral head

## Abstract

Surgical treatment for Legg–Calve–Perthes disease (LCPD) is recommended for older children with moderate to severe disease. We sought to determine whether double osteotomies lead to improved radiologic outcomes compared to reported non-operative outcomes. Patients older than 6 years of age diagnosed with LCPD lateral pillar B or C who were treated with pelvic and femoral osteotomies were included. Radiologic outcomes and leg-length discrepancies were assessed using the Stulberg classification and were compared with the current literature. Fifteen hips in fourteen patients were treated with double osteotomy for LCPD, and seven had lateral pillar C disease (47%). The mean age at surgery was 8.6 years (range, 7.2–10.4) and the mean age at follow-up was 20.2 years (range, 14.2–35.6). At a mean 11.6-year follow-up (range: 6.3–25.2), double osteotomy resulted in 40% of patients having Stulberg I/II scores, 27% having Stulberg III scores, and 33% having Stulberg IV/V scores. The mean leg-length discrepancy was 1.4 cm in lateral pillar C patients compared to 0.8 cm in lateral pillar B patients. Four patients underwent additional surgeries, including two who required total hip arthroplasty. Double osteotomy as an alternative surgical procedure for the treatment of LCPD did not show improved outcomes when compared to historic non-operative cohorts.

## 1. Introduction

Legg–Calve–Perthes Disease (LCPD) is a pediatric hip disorder that can lead to pain, deformity, and functional disability in adulthood [1,2,3,4,5,6]. The etiology of the disease is most likely due to vascular insufficiency [7,8,9], although the complete biological and mechanical mechanisms are not fully understood [10,11]. Both pelvic osteotomy and femoral osteotomy have been described for the treatment of LCPD [12].

Although prior prospective studies have supported surgical management for lateral pillar B or B/C for LCDP [13,14], there are few comparative studies evaluating the role of pelvic vs. femoral osteotomies [15,16,17]. Additionally, there has been limited consensus regarding the best treatment strategy, particularly for patients over 6 years of age and for patients with lateral pillar C disease, where less than 50% of the lateral pillar is maintained [13,14]. In addition, there are drawbacks to both procedures. A femoral osteotomy shortens the limb and causes poor tensioning of the abductors, frequently resulting in a Trendelenburg gait [18]. On the other hand, a pelvic osteotomy is a major surgical procedure that can result in over-coverage and impingement, which is already a frequent occurrence with the coxa magna commonly seen in LCPD. There is one initial report that showed good outcomes with the double-osteotomy approach for LCPD [15]. To date, the use of combined femoral and pelvic osteotomy has been insufficiently studied, especially in patients over age 6 with lateral pillar C disease.

Our institution adopted the double osteotomy in 1997, believing that for LCPD patients diagnosed after age 6 with evident femoral-head lateral pillar involvement (more than 50%), double osteotomy would provide good long-term radiologic outcomes. We sought to evaluate the long-term outcomes of our surgical approach at skeletal maturity.

## 2. Materials and Methods

### 2.1. Study Design

IRB approval was obtained for all aspects of this procedure (12-000263). Patients were included if they were diagnosed with LCPD and treated with double osteotomies between the years of 1997 and 2004. In addition, patients had to have a minimum 5-year follow-up and progress to skeletal maturity (Risser 4 or 5). All patients undergoing surgery had disease onset after age 6 and a diagnosis of lateral pillar B, B/C, or C disease. Non-operative treatment was reserved only for patients with lateral pillar A disease under the age of 6. These patients were excluded from our study. Our search yielded 14 pediatric patients (15 hips), including 10 males and 4 females. The mean age at diagnosis was 7.8 years (range, 6.2–10.0). Patients underwent surgery at a mean age of 8.6 years (7.2–10.4). The mean follow-up time after surgery was 11.6 years (6.3–25.2) and the average age at follow-up was 20.2 years (14.2–35.6) (Table 1).

Hips were graded at the fragmentation stage and classified as either lateral pillar B, B/C border, or C disease [19]. Given our study period, perfusion MRI was not available at the time of diagnosis. The final radiologic outcome was determined by the Stulberg classification [4,13,20,21]. Patients were grouped into either Stulberg I/II, Stulberg III or Stulberg IV/V classifications for final analysis. Three independent physicians graded each of the fifteen hips with a joint review to achieve consensus, and inter-rater reliability statistics were run. In addition to the Stulberg classification, the need for revision surgery and an assessment of leg-length discrepancy on standing radiographs were also analyzed.

### 2.2. Surgical Procedure

Upon diagnosis, eligible patients, based on our inclusion criteria, underwent combined femoral and pelvic osteotomies. The femoral osteotomy (opening wedge, single osteotomy perpendicular to the long axis of the femoral shaft at the intertrochanteric level) was performed, aiming for 15 degrees of correction into the varus. This was assessed intraoperatively with triangular wedges and compared to our preoperative plan. A Salter osteotomy was performed after the femoral osteotomy was fixed, with the degree of correction guided by intraoperative fluoroscopy. The goal of the procedures was to achieve a contained femoral head with complete coverage of the lateral pillar under the bony acetabulum. Patients were instructed to be toe-touch or non-weightbearing for 6–12 weeks until the bony union was achieved. Prolonged non-weightbearing was not part of the treatment protocol. Patients routinely underwent implant removal 1 year following surgery.

### 2.3. Analysis

Differences in leg length at maturity between lateral pillar B and C patients were analyzed using an independent *t*-test. Statistical analyses were performed using BlueSky Statistics. Significance was set at α = 0.05. Our data were then compared to the current literature.

## 3. Results

There were eight patients (53%) with lateral pillar B disease and seven patients (47%) with lateral pillar C disease (Table 1 and Table 2). At final follow-up, 6 (40%) of the 15 hips were classified as Stulberg II, 4 (27%) were classified as Stulberg III, and 5 (33%) were classified as Stulberg IV/V. Three (37.5%) of the eight pillar B hips were classified as Stulberg II, three (37.5%) were Stulberg III, and two (25%) were Stulberg IV/V. In contrast, three (43%) of the seven pillar C hips were classified as Stulberg II, one (14%) was Stulberg III, and three (43%) were Stulberg IV/V. Three of the eight hips in the lateral pillar B group were spherical (Stulberg I or II) and three of the seven hips in the lateral pillar C group were spherical. Inter-rater reliability calculations showed excellent agreement, with an ICC of 0.91.

Thirteen of the fourteen patients had residual leg-length discrepancy, as measured on the latest standing anteroposterior pelvis radiographs. The average leg-length difference was 1.1 cm (Table 2). Furthermore, average leg-length discrepancy also differed by lateral pillar classification. Patients with lateral pillar B disease had an average leg-length discrepancy of 0.8 cm while patients with lateral pillar C disease had an average leg-length discrepancy of 1.4 cm; however, this difference was not statistically significant (*p* = 0.64). Only two patients in the cohort had discrepancies of over 2 cm.

No perioperative complications were observed. Four of the fourteen patients (28.6%) have undergone a total of six revision reconstructive surgeries. One patient underwent a left-hip surgical dislocation, trochanteric advancement, and labral repair 13 years after her initial procedure. Six months later, the same patient had a periacetabular osteotomy. Another patient underwent left distal femoral epiphysiodesis 5 years after the initial procedure and then had a subsequent right-hip offset procedure, labral repair, and neck lengthening 5 years after that. A third patient had a revision of the right proximal femoral valgus-producing and internal rotation-producing osteotomy 5 years after the initial procedure, before undergoing total hip arthroplasty (THA). The final patient underwent THA 25 years after the initial osteotomy, yielding a 14.3% conversion to THA (Figure 1).

## 4. Discussion

Earlier studies have supported the use of osteotomies in children older than 6 to 8 years old, resulting in improved Stulberg classification scores at follow-up compared to their non-operative counterparts [13,14,22]. Two high-quality prospective studies show improved radiographic outcomes for the treatment of lateral pillar B and B/C hips with surgical management for patients over the age of 6 to 8. However, they found that the surgery must be performed early in the disease process [12,13]. Wiig et al. studied femoral osteotomy vs. non-operative management for the treatment of patients over age 6 [13]. Herring et al. compared operative vs. non-operative cohorts and included both femoral and pelvic osteotomies in their prospective multicenter study [12]. Both studies found that operative management improved the Stulberg rating at skeletal maturity in patients older than 6 to 8 who are early in the disease process [13,14]. In addition, surgery has not been shown to provide short-term health benefits such as improved health-related quality of life or patient function in childhood.

Interestingly, in a 20-year follow-up, a cohort of patients treated non-operatively had only a 5% rate of total hip arthroplasty at 20 years and reasonable outcome scores, with a mean non-arthritic hip score of 79, although 75% of patients had residual hip pain [23]. Only Stulberg I/II patients had low rates of osteoarthritis (at 22%), while Stulberg III and IV/V had rates of osteoarthritis of 61% and 62%, respectively, suggesting that only spherical hips do well at long-term follow-up. While our study reported a larger percentage of patients with lateral pillar C disease, our results indicated worse outcomes, with a 20% rate of revision surgery and 7% THA at a follow-up of just over 10 years. Of the revisions in our series, two had lateral pillar C disease and one had lateral pillar B (Figure 2).

Biomechanical analyses of femoral and pelvic osteotomies have been conducted to assess the amount of coverage and force placed on the femoral head, both of which may affect treatment success. Rab looked at the differences between these two procedures and found better anterior coverage with pelvic osteotomies than with femoral osteotomies [24]. Furthermore, Rab’s analysis of forces on the hip after pelvic osteotomy and femoral osteotomy showed that the pelvic osteotomy allowed for a more physiologic distribution of force during the gait cycle [24]. Conversely, the femoral osteotomy produced a greater medial sheering force on the epiphyseal plate [24]. However, several studies report that pelvic osteotomies can cause acetabular retroversion [11,25,26,27,28], which is a risk factor for impingement compounding the femoral-head deformity seen with LCPD [23,29].

Single osteotomy is the most common surgical approach for the treatment of LCPD [13,15,30,31]. Previous studies have compared the results of femoral and pelvic osteotomies. Sponseller et al. compared the radiographic findings of children who had undergone either femoral or pelvic osteotomy at a mean 9-year follow-up [31]. In this study, 18 (69%) of the femoral osteotomy patients and 22 (65%) of the pelvic osteotomy patients had a final Stulberg classification score of I or II [31]. Similarly, Kitakoji et al. found that femoral de-rotational osteotomies and pelvic osteotomies had similar results at a roughly 10-year follow-up [15]. In both surgical groups, four (40%) had a Stulberg classification score of I or II, three (30%) had a Stulberg classification score of III, and three (30%) had poor Stulberg classification scores (IV) [15].

Previous reports on the results of double osteotomy for LCPD have been mixed [17,18]. Mosow et al. reported on 52 children who had undergone double osteotomies [18]. At the mean 10-year follow-up, 27 patients (51%) had a Stulberg I/II result, 15 (29%) had a Stulberg III result, and 10 (20%) had a Stulberg IV/V result [18], which are better than our results. Lim et al. conducted a similar study on 12 LCPD patients older than 8 who received double osteotomies, with a mean 10-year follow-up [32]. At the final follow-up, four patients (33%) had a Stulberg classification score of II, seven (58%) had a Stulberg classification score of III, and one (8%) patient had a Stulberg classification score of IV [32]. Similar small series have reported mixed results with double osteotomies [33,34].

Our study results were similar to those reported with non-operative management. Herring et al. reported on children over the age of 8 treated non-operatively [13]. Of the 91 non-operative patients, 30 (33%) had a final Stulberg result of I or II, 34 (37%) had a Stulberg III result, and 27 (30%) patients had a Stulberg IV or V result [13], which are similar to the results we reported with double osteotomies. Long-term results from the same cohort at a 20-year follow-up showed 23 patients (40%) with a Stulberg I/II result, 19 patients (33%) with a Stulberg III result, and 15 patients (26%) with a Stulberg IV/V result [23], although 75% of patients reported hip pain. This is nearly identical to our cohort of fifteen patients treated with a double osteotomy at a mean follow-up of 12.5 years, with six patients (40%) having Stulberg I/II classification, four patients (27%) having Stulberg III classification, and five patients (33%) having a Stulberg IV/V classification (Table 3).

Leg-length discrepancies and gait abnormalities were common in this cohort. Our double-osteotomy patients were left with a mean 1.1 cm leg-length discrepancy, which did not differ by lateral pillar group. Previous studies have shown that the amount of varus angulation at the time of osteotomy did not correlate well with radiographic results [31]. The lengthening achieved by the pelvic osteotomy did not fully correct for the leg-length discrepancy resulting from LCPD and varus osteotomy.

Our results have several limitations, including a higher proportion of lateral pillar C patients (50%) than previous series and small patient numbers. This higher proportion of lateral pillar C patients in our cohort may be reflective of our referral patterns as a tertiary referral center in the area. Our series did not evaluate the results of surgical treatment for patients under age 6 at disease presentation. In addition, the time from disease onset to intervention may have an impact on patient outcomes. A larger dataset may allow for further analysis of outcomes based on the Waldenstrom classification at the time of intervention [35,36]. This study did not address health-related quality of life and rates of subsequent arthritis or the need for total hip arthroplasty or other potential benefits of surgical management. An additional long-term follow-up of these patients is needed to assess the success of these procedures in the context of conversion to total hip arthroplasty, impingement, and arthritis. The patients in our series were compared to controls over age 6 in the literature (Table 3). Because we excluded patients 6 years and under, complete data such as age at surgery were not available on the literature controls. Nevertheless, it is also important to consider the psychosocial burden of the procedure on the family and the child. A double osteotomy is an extensive surgery that requires substantial parental aftercare and is difficult for a young child. Additionally, a second surgery for implant removal is typically required. Non-operative treatment may involve prolonged bracing or limited weightbearing, which has also been shown to have a negative social and physical impact on patients with LCPD [37].

In summary, the long-term outcomes from a double osteotomy for the treatment of LCPD may be limited and potentially not improved compared to historical non-operative cohorts. Future approaches may include pharmacologic and biologic interventions to improve the local healing response in this challenging disease process.

## Figures and Tables

**Figure 1 jcm-12-05718-f001:**
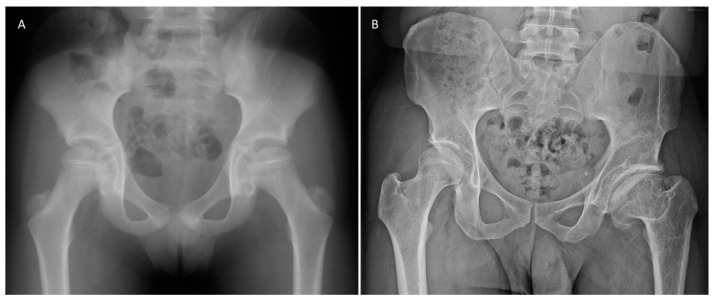
(**A**). Initial presenting radiograph at age 9 for patient who eventually developed lateral pillar C disease. (**B**). Patient at most recent follow-up to plan for THA at age 36. Stulberg score of IV was given.

**Figure 2 jcm-12-05718-f002:**
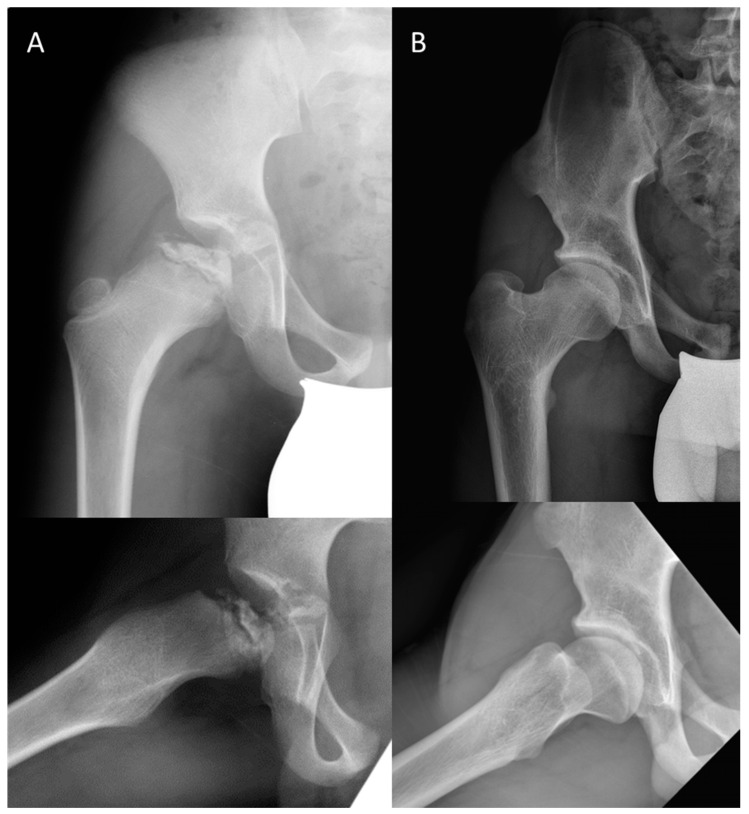
(**A**). Patient with lateral pillar C disease who presented at age 8. (**B**). Patient at most recent follow-up at age 18; although patient has a spherical head (Stulberg II), the high-riding greater trochanter resulted clinically in abductor insufficiency.

**Table 1 jcm-12-05718-t001:** Demographics.

Patient Number	Gender	Lateral Pillar	Age at Diagnosis (Years)	Age at Surgery (Years)
1	M	C	8.3	9.2
2	M	B	6.4	8.3
3	F	C	7.0	7.2
4	F	C	6.5	7.6
5	M	C	9.0	10.4
6	M	B	7.0	8.3
7	F	B	8.0	8.6
8	M	B	9.0	9.1
9	M	C	10.0	10.4
10	M	B	7.3	7.9
11	M	C	7.7	8.7
12	M	C	6.8	7.4
13	F	B	9.6	9.7
14 (R)	M	B	7.0	7.3
14 (L)		B		
Group Mean ± SD			7.8 ± 1.2	8.6 ± 1.1

R: right hip, L: left hip; db-osteotomy: combined proximal femur and Salter osteotomies.

**Table 2 jcm-12-05718-t002:** Radiologic outcomes after skeletal maturity.

Patient #	Stulberg	Leg-Length Discrepancy (cm)	Age at Latest Visit (Years)	F/U from Diagnosis (Years)	F/U from Surgery (Years)
1	II	0.2	16.1	7.7	6.9
2	II	1.5	14.7	8.3	6.3
3	V	1.8	27.0	20.0	19.8
4	III	0.5	26.2	19.7	18.6
5	IV	3.9	35.6	26.6	25.2
6	III	2.4	22.4	15.4	14.1
7	II	0.3	17.7	9.7	9.1
8	III	0.2	15.9	6.9	6.8
9	IV	1.1	23.6	13.6	13.1
10	II	0.6	14.2	6.9	6.3
11	II	1.2	18.3	10.6	9.6
12	II	1.0	18.1	11.3	10.7
13	IV	0.8	18.2	8.7	8.6
14 (R)	IV	0.0	14.6	7.6	7.4
14 (L)	III				
Group Mean ± SD		1.1 ± 1.0	20.2 ± 6.1	12.3 ± 6.0	11.6 ± 5.9

**Table 3 jcm-12-05718-t003:** Results from the literature for patients over age 6.

Study	Procedure	Lateral Pillar C Patients	Stulberg I/II	Stulberg III	Stulberg IV/V	Total
Ours	Double Osteotomy	7 (47%)	6 (40%)	4 (27%)	5 (33%)	15
Herring et al. [13]	Single Osteotomy	5 (10%)	25 (50%)	16 (32%)	9 (18%)	50
Shohat et al. [30]	Single Osteotomy	3 (19%)	7 (50%)	7 (50%)		14
Sponseller et al. [16]	Femoral Osteotomy	not reported	18 (69%)	8 (31%)		26
	Pelvic Osteotomy	not reported	22 (65%)	12 (35%)		34
Kitakoji et al. [15]	Femoral Osteotomy	26 (56%)	4 (40%)	3 (30%)	3 (30%)	10
	Pelvic Osteotomy	18 (60%)	4 (40%)	3 (30%)	3 (30%)	10
Mosow et al. [18]	Double Osteotomy	28 (68%)	27 (51%)	15 (29%)	10 (20%)	52
Lim et al. [32]	Double Osteotomy	12 (100%)	4 (33%)	7 (58%)	1 (8%)	12
Javid and Wedge [33]	Double Osteotomy	2 (10%)	6 (30%)	9 (45%)	5 (25%)	20
Crutcher and Staheli [34]	Double Osteotomy	not reported	7 (50%)	6 (43%)	1 (7%)	14
Herring et al. [13]	Non-operative	14 (15%)	30 (33%)	34 (37%)	27 (30%)	91
Larson et al. [23]	Non-operative	11 (19%)	23 (40%)	19 (33%)	15 (26%)	57

## Data Availability

Data available upon request.

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
