# Peer review of "Long-Term Outcomes at Skeletal Maturity of Combined Pelvic and Femoral Osteotomy for the Treatment of Legg–Calve–Perthes Disease"

_jcm, 2023, doi:10.3390/jcm12175718_

Round 1
Reviewer 1 Report
Treatment in LCPD over 8 years has a bad prognosis in general. In this respect, the rate of 20% revision surgery and 7% THA is not surprising.
Unfortunatly, a large number of participants (50%) had lateral pillar C and was thus in the fragmentation stage. An additional point for the discossion would be if an early intervention before fragmentation (Waldenström stage 1) would lead to better results as studies showed (Joseph 2006 and Sankar 2020)
Another interesting point for the discussion is the postop loading. How long patients did not weight bearing or when patient went back to full weight bearing postop.
Author Response
Thank you so much for your suggestions. We have added, 2nd to last paragraph: “In addition, time from disease onset to intervention may have an impact on patient outcomes. A larger dataset would allow for further analysis of outcomes based on Waldenstrom classification at time of intervention [37]. “
Patients were only non-weightbearing until the osteotomies had healed. We have clarified this point in the methods.
“Patients were instructed to be toe-touch or non-weightbearing for 6-12 weeks until bony union was achieved. Prolonged non-weightbearing was not part of the treatment protocol.”
Reviewer 2 Report
1. Table II requires age at the initial diagnosis since age also plays role
2. Line 161. sentence error. "spherical hips do well at long term follow-up. in the long term. While our study"
3. Line 186 and 203, Need to spell the number if it is at the beginning of the sentence.
4. Table 3. Need to add what was the mean age when the procedure was performed
5. Add radiograph of lateral pillar B with Stulberg IV/V outcome as comparison, showing that even with pillar B, Stulberg IV/V outcome can still happen
6. remove the vertical borders of the table, leaving only the horizontal lines
7. The authors are less discussing about the influence of age on the Stulberg outcome. What would be the authors' approach if the child already had pillar C but the age is slightly younger than those children with pillar B?
8. Authors need to recommend wisely based on their study, based on the earlier studies as well as based on their experience, which approach will be wiser especially for young pediatric surgeons upon managing LCPD.
Author Response
- Table II requires age at the initial diagnosis since age also plays role
Thank you so much for your suggestion. The initial age at diagnosis is in the first table.
- Line 161. sentence error. "spherical hips do well at long term follow-up. in the long term. While our study"
Thank you so much for your comments. The changes have been made.
- Line 186 and 203, Need to spell the number if it is at the beginning of the sentence.
We have changed these sentences.
In this study, 18 (69%) of the femoral osteotomy patients and 22 (65%) of the pelvic osteotomy patients had a final Stulberg classification score of I or II [31].
Herring et al. reported on children over the age of 8 treated nonoperatively [13]. Of the 91 non-operative patients, 30 (33%) had a final Stulberg result of I or II, 34 (37%) had a Stulberg III, and 27 (30%) patients had a IV or V [13], which is similar to our results reported with double osteotomies.
- Table 3. Need to add what was the mean age when the procedure was performed
We only analyzed patients that were older than 6 from these studies. The average age of these smaller cohorts is unknown. We added the following sentences to the 2nd to last paragraph/limitations paragraph.
The patients in our series were compared to literature controls over age 6 (Table 3). Because we excluded patients 6 years and under, complete data such as age at surgery was not available on the literature controls.
- Add radiograph of lateral pillar B with Stulberg IV/V outcome as comparison, showing that even with pillar B, Stulberg IV/V outcome can still happen
We have added the lateral images.
- remove the vertical borders of the table, leaving only the horizontal lines
We have made those changes.
- The authors are less discussing about the influence of age on the Stulberg outcome. What would be the authors' approach if the child already had pillar C but the age is slightly younger than those children with pillar B?
It is our practice to offer surgery to patients with severe disease (lateral pillar B, B/C, C or now greater than 50% involvement of the femoral head on perfusion MRI). We do not offer surgery for children under age 6. Surgery under age 6 has not been shown to make a difference in treatment.
- Authors need to recommend wisely based on their study, based on the earlier studies as well as based on their experience, which approach will be wiser especially for young pediatric surgeons upon managing LCPD.
Thank you. We made recommendations for the treatment of children greater than age 6 based on the data collected. We understand that we do not have the comprehensive data needed to make decisions for all patients.
We added this sentence to the limitations paragraph.
Our series did not evaluate the results of surgical treatment for patients under age 6 at disease presentation.